# DNA Methylation Analysis Identifies Patterns in Progressive Glioma Grades to Predict Patient Survival

**DOI:** 10.3390/ijms22031020

**Published:** 2021-01-20

**Authors:** Jing Yin Weng, Nicole Salazar

**Affiliations:** Department of Biology, San Francisco State University, San Francisco, CA 94132, USA; jweng2@mail.sfsu.edu

**Keywords:** glioma, glioblastoma, DNA methylation, progression, TCGA, WGCNA, prognosis

## Abstract

DNA methylation is an epigenetic change to the genome that impacts gene activities without modification to the DNA sequence. Alteration in the methylation pattern is a naturally occurring event throughout the human life cycle which may result in the development of diseases such as cancer. In this study, we analyzed methylation data from The Cancer Genome Atlas, under the Lower-Grade Glioma (LGG) and Glioblastoma Multiforme (GBM) projects, to identify methylation markers that exhibit unique changes in DNA methylation pattern along with tumor grade progression, to predict patient survival. We found ten glioma grade-associated Cytosine-phosphate-Guanine (CpG) sites that targeted four genes (*SMOC1, KCNA4, SLC25A21*, and *UPP1*) and the methylation pattern is strongly associated with glioma specific molecular alterations, primarily isocitrate dehydrogenase (IDH) mutation and chromosome 1p/19q codeletion. The ten CpG sites collectively distinguished a cohort of diffuse glioma patients with remarkably poor survival probability. Our study highlights genes (*KCNA4* and *SLC25A21*) that were not previously associated with gliomas to have contributed to the poorer patient outcome. These CpG sites can aid glioma tumor progression monitoring and serve as prognostic markers to identify patients diagnosed with less aggressive and malignant gliomas that exhibit similar survival probability to GBM patients.

## 1. Introduction

DNA methylation is a heritable epigenetic marker in the genome that does not impact the genetic makeup through the addition of a methyl group to the DNA molecule. Most DNA methylation in humans occur at the 5′ carbon of the cytosine base which is followed by a guanine nucleotide. Addition of the methyl group changes the chromatin structure, making it more condensed, which results in DNA being less accessible for transcription [1,2]. DNA methylation patterns change during development [3] and with age [4]. DNA methylation in the genome also exhibits tissue- and cell-type differences, for example, highly methylated content is observed in brain tissue [5]. Other than these differences, which occur naturally during the human life cycle, methylation changes are also related to diseased cell states. Many studies have revealed two common alterations in cancer: DNA hypermethylation and hypomethylation [6,7,8]. Hypermethylation involves DNA methylation events at the Cytosine-phosphate-Guanine (CpG) site within the promoter of tumor suppressor gene while hypomethylation removes the methyl group at the promoter site of oncogenes and can be widespread around the genome to promote tumor progression [9,10].

Although the genome is the same in essentially all cells of the human body, the epigenome differs from cell to cell and is dynamic, changing with time and exposure to the environment. Epigenetic mechanisms affect all steps of the gene expression process, from chromatin state to transcription, post-transcriptional RNA processing, and translation. Epigenetic mechanisms for regulating gene expression include DNA methylation and histone modification, arguably the best-established epigenetic processes that regulate gene expression at the level of transcription. Histones are highly conserved basic proteins around which DNA is tightly wrapped to form nucleosomes, or the “beads on a string structure” that make up chromatin. Histone tails protrude from the nucleosome and are subject to a variety of post-translational modifications (PTMs), primarily methylation, acetylation, and phosphorylation. These PTMs affect gene expression by controlling accessibility of the chromatin structure to expose DNA-binding sites or by closing DNA-binding sites to facilitate transcription [11]. Non-coding RNAs also have a role in epigenetic modification by regulating gene expression and chromosomes to control cell differentiation [12]. In addition, gene expression can be regulated post-transcriptionally through dynamic and reversible RNA modifications. Extra methyl group modifications induce either duplex stability or protein–RNA affinity and positively correlate with translation [13]. Epigenetic mechanisms for regulating gene expression clearly are complex and diverse to create a dynamic epigenome and epitranscriptome.

Glioma is a type of cancer that occurs in the brain and it is the most prevalent brain tumor observed in the population. Gliomas are comprised of four subtypes which are classified based on the World Health Organization of Central Nervous System Tumor Classification guidelines updated in 2016, with the introduction of molecular biomarkers to further refine the diagnostic criteria [14]. Grade II and Grade III gliomas include astrocytoma, oligodendroglioma, and oligoastrocytoma which are frequently considered to have significantly better patient survival than Grade IV glioma [15]. Glioblastoma multiforme (GBM), a Grade IV glioma subtype, is an aggressive brain tumor and one of the most fatal due to its lack of curable treatment. The median survival time for patients is between 15 to 16 months [16]. One commonly observed biological event that affects patient survival time is DNA methylation [17]. Studies confirmed a phenomenon called CpG Island Methylator Phenotype (CIMP) after analyzing DNA methylation data of GBM and LGG (lower glioma grade) cohorts from The Cancer Genome Atlas (TCGA) database. It noted hypermethylation on the CpG island of a subset of genes, including *ANKRD43*, *HFE*, *MAL*, and *FAS-1*, and classified tumor samples as CIMP-positive or CIMP-negative based on the methylation level detected on those genes. Glioma patients with CIMP-positive tumors correlate with better survival. Another well-studied methylation biomarker in GBM patients is in the DNA repair gene called O6-methylguanine–DNA methyltransferase (*MGMT*). Methylation on the promoter site of *MGMT* reduces gene expression and protein activity to prevent it from rescuing tumor cells with alkylating agent-induced damage caused by chemotherapy [18]. As a result, GBM patients with loss of or low MGMT activity have higher sensitivity to temozolomide, a common chemotherapy used to treat GBM. Although there is growing attention around the classification of glioma patients through molecular profiling such as genetic and methylation signatures [19], studying the connection between DNA methylation pattern changes and tumor grades, which is classified through histological characteristics as a standard protocol by the World Health Organization, will improve knowledge about glioma tumor progression.

Despite the current understanding around how alterations in DNA methylation induce tumor generation and progression [20,21,22] or predict patient outcome independently [23,24,25], it is unclear how DNA methylation changes progress between tumor grades or how they influence patient survival. We hypothesized there is a correlation between methylation pattern and glioma grade and such correlation could be used for tumor progression monitoring as well as patient prognosis. We evaluated the changes in DNA methylation patterns in LGG and GBM patients and identified changes in methylation level for CpG sites that have significant impact on patient survival.

## 2. Results

### 2.1. Differentially Variable Cytosine-phosphate-Guanine Sites Identified through Sample Comparison of Different Glioma Grade

The training data set was comprised of 400 glioma samples (Table 1) and β-values, the methylation level of 180,758 CpG sites that were captured through the probes designed in Infinium HumanMethylation450K beadchip. We performed two comparisons to assess the methylation changes between each glioma grade and its subsequent higher tumor grade: Grade II vs. Grade III and Grade III vs. Grade IV. In the comparison between Grade II and Grade III gliomas, 2095 and 146 CpG sites (1.54 × 10^−10^ ≤ *p*-value ≤ 0.00062, 2.75 × 10^−5^ ≤ false discovery rate (FDR) ≤ 0.05) were consistently hypomethylated or hypermethylated, respectively, in one glioma grade while the other glioma grade shows variability (Figure 1A). The comparison between Grade III and Grade IV gliomas returned 28,503 hypomethylated and 30,731 hypermethylated CpG sites (3.38 × 10^−61^ ≤ *p*-value ≤ 0.016385, 6.11 × 10^−56^ ≤ FDR ≤ 0.05). The remaining CpG sites that are not part of the hypo- and hypermethylated groups were considered insignificant in the differential variability analysis with FDR > 0.05. Differential variability analysis output can be found in Appendix A. We selected all the significant CpG sites from the Grade II and Grade III comparison and a subset of significant CpG sites (top 10% by FDR) from the Grade III and Grade IV comparison for visualization of the methylation expression (Figure 1B) in the form of heatmaps. Although the number of hyper- and hypomethylated CpG sites are comparable in the Grade III and Grade IV comparison, more hypomethylated CpG sites were selected for heatmap visualization due to the smaller FDR.

### 2.2. Comethylation Modules Displaying Correlation with Glioma Grade Progression Identified through Weighted Correlation Network Analysis

We used the weighted [gene] correlation network analysis (WGCNA) to find methylation modules of highly correlated CpG sites [26]. Of the 59,234 significant DV CpG sites detected between Grade III and Grade IV glioma comparison, the top 25% were selected based on the lowest FDR. Together with the 2241 significant DV CpG sites from the Grade II and Grade III comparison, we ran a total of 16,244 unique DV CpG sites through the hierarchical clustering method in the network construction function. In Figure 2A, we show the cluster dendrogram for one of the data blocks consisting of approximately 5000 DV CpG sites used during network construction. WGCNA generated eight comethylation modules, as depicted in Figure 2B, where each color represents one comethylation module with module size shown on the legend. CpG sites without significant correlation with any of the eight modules were assigned to the grey module and excluded from the pie chart.

To identify methylation changes in CpG sites that contribute to tumor progression, we correlated the comethylation modules to the glioma grade (Figure 2C). The brown, turquoise, and blue methylation modules showed correlation greater than |0.5|. The CpG sites from the brown and turquoise modules exhibited a negative correlation with tumor grade (cor = −0.57, *p*-value = 2 × 10^−35^ and cor = −0.61, *p*-value = 2 × 10^−40^, respectively) indicating methylation level decreased as tumor grade advanced. On the other hand, the blue methylation module had a positive correlation (cor = 0.54, *p*-value = 2 × 10^−30^) indicating the methylation level of the CpG sites increased along with tumor grade.

### 2.3. Candidate Genes and CpG Sites that Contribute to Tumor Progression

To evaluate the methylation pattern changes associated with tumor progression, CpG sites with low correlation to glioma grade were eliminated and the analysis continued with CpG sites that have significance ≥ 0.5 from the brown, turquoise, and blue module with trait significance ≥ |0.5|. Figure 3A shows the correlation of the module membership and CpG site significance of individual CpG sites. CpG site significance measures the biological significance of the CpG sites to glioma grade progression. Module membership evaluates the correlation between the methylation profile of a given CpG site with the remaining CpG sites of the module. The three significant modules have a high correlation between CpG site significance and module membership indicating the CpG sites with high significance contribute substantially to module and trait relationships. This filter reduced the CpG sites to 3551, which distributed across 2107 genes (Figure 3B). We increased our chance of identifying tumor progression-associated genes and the corresponding methylation sites by including only genes with 5 or more CpG sites associated with them. This step disqualified most of the genes and retained 150 candidate genes for the next analysis.

### 2.4. Establishing Correlation between Gene Expression, Methylation Level, and Glioma Grade

We evaluated the methylation profile of the 150 candidate genes on the associated significant CpG sites identified from the previous step through heatmap visualization (Appendix A) and observed that a subset of the genes exhibited steady methylation change along with tumor grade advancement. We also examined the gene expression of the candidate genes through boxplots (Appendix A). Figure 4A shows gene expression of four genes that showed either an increase (*UPP1*) or reduction (*SMOC1*, *KCNA4*, and *SLC25A21*) in gene expression as tumor grade advances. Together with the methylation heatmaps of the genes, we found the corresponding methylation CpG sites that may have contributed to the regulation of gene expression and plotted the methylation level of the training samples, by glioma grade, for confirmation (Figure 4B). There was a strong correlation between gene expression and methylation level at some specific CpG sites of the gene; the beta-value of the CpG sites for *SMOC1*, *KCNA4*, and *SLC25A21* increases as glioma grade progresses and results in decreased gene expression and the opposite trend was observed for *UPP1*. To confirm the validity of our observation, we examined gene expression and methylation level of the *IGF2BP3* and *Fn14* genes as previous studies have demonstrated that higher gene expression is associated with higher glioma grade and negatively correlate with methylation level (Figure 4C) [22,27]. In addition, our results obtained from the training data set are consistent with those studies; patients with high gene expression of *IGF2BP3* and *Fn14* correlate with low methylation levels and worse survival.

### 2.5. Discovery and Validation of Prognosis Markers Associated with Tumor Progression

After establishing the correlation between gene expression and methylation level of the candidate genes, CpG sites with a strong negative correlation with the corresponding genes were next used for survival analysis. Figure 5 shows the Kaplan-Meier plots of CpG sites illustrating the survival analysis. All training samples were separated into two groups by comparing their methylation level for the CpG site of interest to the methylation median of the training set: hypermethylation and hypomethylation. Ten CpG sites targeting *SMOC1*, *KCNA4*, *SLC25A21*, and *UPP1* genes, demonstrated significant survival differences between the two groups, and nine of them are presented in the figure. A summary of the ten CpG sites is given in Table 2.

### 2.6. Establishing a Prognostic Model from the Methylation Markers

The last step of the study was to explore all ten methylation markers to check if any of them contradict each other when used for patient prognosis collectively. Figure 6 shows a schematic diagram illustrating how the samples are separated into good and poor prognosis groups during survival analysis. We assigned the training set of patients with methylation profiles that met the poor patient survival for all ten CpG sites to the poor prognosis group, which was a total of 70 patients, and the remaining 330 patients as the good prognosis group. The survival probability of the poor prognosis group dropped dramatically with a median survival time of 414 days compared to 2433 days for the good prognosis group (Figure 7). This set of CpG sites are validated with the validation sample set described in Table 1 and the patients showed a similar trend in prognosis; median survival time was 544 days for the poor prognosis group while the good prognosis group was at 2235 days. Patient distribution, by glioma grade, of the good prognosis and poor prognosis group for both sample sets is provided in Table 3.

### 2.7. Correlation of Prognostic Markers to Clinical and Molecular Data

To assess the relationship between the methylation pattern of the ten prognostic CpG sites and tumor molecular features, we plotted the methylation data in heatmaps (Figure 8A) with glioma grade progression and other molecular alterations that are frequently observed in gliomas for the training samples. cg16270885, the probe that targets the CpG site of the *UPP1* gene shows a decrease in methylation signal as tumor grade advances while the rest of the CpG sites were methylated as tumor grade progressed. We assessed the correlation of our prognostic markers with the molecular subtypes described by Ceccarelli et al. [19] and the methylation pattern of these CpG sites are similar for samples within the same molecular subtype; the CpG sites we identified align with the molecular profile that distinguishes glioma samples into smaller subtypes suggested by Ceccarelli et al. The effect of isocitrate dehydrogenase (IDH) and chromosome 1p/19q codeletion alterations on these ten prognostic markers was also evaluated and cg16270885 shows a sharp methylation pattern between different IDH and 1p/19q codeletion status. The highest methylation signal is observed in samples with IDH mutation and chromosome 1p/19q codeletion, but the methylation signal slightly decreased in samples with IDH mutation alone. The methylation signal of the IDH wild type and no codeletion glioma samples was the lowest compared to the other two groups. Validation samples show similar patterns as the training samples (Appendix A).

Sample distribution within the different prognostic groups is summarized in Table 3 with clinical and molecular data presented in bar charts in Figure 8B. All samples in the poor prognostic group have an IDH wild type genotype and nearly all belong to classic-like or mesenchymal-like molecular subtypes. Patients with poor outcomes are made up primarily of Grade IV GBM and Grade III astrocytoma or Grade III oligoastrocytoma. A small fraction of the poor prognosis samples is oligodendroglioma from either Grade II or Grade III. Moreover, a large portion of poor patient outcome associated with copy number alteration (chromosome 7 amplification and 19 deletion) and the corresponding samples were classified as classic-like or mesenchymal-like molecular subtypes by Ceccarelli et al.

## 3. Discussion

With differential variability analysis, we identified CpG sites that behave similarly within the same glioma grade but differently from the other glioma grades. We compared Grade II vs. Grade III and Grade III vs. Grade IV and identified 2241 and 59,234 significantly differentially variable (DV) CpG sites, respectively. The epigenetic profile difference between Grade III and Grade IV was much greater than the differences observed between Grade II and Grade III. It was not surprising since previous methylation profiling studies have shown a similar pattern where Grade II and Grade III samples are more likely to be clustered together and separate from Grade IV [19,22]. We noticed a progressive demethylation condition in the top 10% of the DV CpG sites via the methylation profiling heatmap and this observation aligns with the observations from other studies [22,28].

We applied the network construction function from WGCNA to the subset of CpG sites and found three comethylation modules highly correlated with the progression of glioma grade, the clinical trait of interest in our study. Most of the CpG sites displayed a negative correlation with the increase of tumor grade indicating that more CpG were demethylated as the tumor progresses. The demethylation pattern in the data set indicates the up-regulated gene expression in higher glioma grade samples. As hyper- and hypomethylation in cancer normally applies to extensive methylation or demethylation around the promoter site, we assessed genes that have five or more significant CpG sites associated with them. Out of the 150 candidate genes, four of them, *SMOC1*, *KCNA4*, *SLC25A21*, and *UPP1* were most outstanding due to the distinct gene expression pattern changes between the three glioma grades. Gene expression of *SMOC1*, *KCNA4*, and *SLC2521* declined as patient glioma grade increased while *UPP1* gene expression showed a positive relationship with tumor grade. Ten of the CpG sites associated with these four genes displayed a strong inverse relationship between gene expression and methylation level implying a high probability that these CpG sites impact gene expression.

*SMOC1* gene encodes a matricellular protein called secreted modular calcium-binding protein 1 [29] and this protein was shown to regulate growth factors [30,31,32]. Boon et al. showed *SMOC1* is a Grade II and III astrocytoma-associated gene [33] and this conclusion aligns with the gene expression data we analyzed; the greatest median gene expression was observed in Grade II glioma samples and subsequently dropped particularly low for Grade IV glioma samples. Previous studies revealed several functions of SMOC1 including the promotion of angiogenesis through regulation of transforming growth factor β signaling pathway in cultured endothelial cells [32] and inhibition of cell migration induced by tenascin-C, an extracellular protein that is overexpressed in many human cancer types, in glioma cell lines [34]. Our analyzed data showed a dramatic reduction in *SMOC1* expression and we predict the main function of *SMOC1* in Grade IV glioma is to promote tumor invasion and migration since rapid spreading is one of the signatures of glioblastoma. Our survival analysis of the *SMOC1* targeted CpG sites reflected a poor prognosis in highly methylated samples which corresponded to down-regulation of *SMOC1* gene expression.

KCNA4 (Potassium voltage-gated channel subfamily A member 4, aka Kv1.4) is a member of the potassium voltage-gated channel family and one of its major functions is the cardiac transient outward K(+) currents [35]. However, Zheng et al. observed hypermethylation at *KCNA4* promoter site in serum as well as tissue samples of gastric cancer patients, and it was one of the markers which showed good sensitivity and specificity for detection [36]. Coma et al. found a global reduction in voltage-gated potassium channel expression, including *KCNA4*, in the brain of tumor-bearing animals suffering from cancer cachexia [37]. Although there is limited information around how *KCNA4* is associated with tumors, other members from the potassium voltage-gated channel family such as Kv1.3 and Kv1.5 are well-studied and have shown correlations with several human cancers including gliomas [38,39]. Further investigation on how *KCNA4* expression and methylation impact glioma is needed. The *KCNA4* CpG sites we identified indicate worse patient survival in the hypermethylated group correlating with decreased gene expression as glioma grade progresses.

Solute Carrier Family 25 Member 21 is encoded by the *SLC25A21* gene and it catalyzes the transportation of 2-oxoadipate and 2-oxoglutarate across the mitochondrial membranes [40]. Rochette et al. reviewed several abnormal *SLC25* activities that are linked to cancer including the overexpression of *SLC25A1* in lung cancer, *SLC25A43* gene deletion as well as elevated *SLC25A33* expression in breast cancer, and SLC25A10 that regulates the redox homeostasis was also increased in multiple cancer types [41]. More importantly, *SLC25A12* expression in hepatocellular carcinoma cell line increased through the modification of histone acetylation [42]. Although the impact of *SLC25A21* on cancer has not been evaluated, the vast number of studies done on other members of the SLC25 family implies that *SLC25A21* can also be a promising tumor-associated marker. This is proved with the survival analysis of our study where methylation changes observed in different glioma grade serves as a good prognosis marker; hypermethylation on the CpG site targeted by cg25051529 is associated with poorer survival.

*UPP1* gene encodes the Uridine Phosphorylase 1 that catalyzes the reversible phosphorylation of uridine to uracil [43] and maintains uridine homeostasis [44]. Overexpression of the *UPP1* is associated with cancer. Guan et al. analyzed the TCGA cohort and found elevated *UPP1* in thyroid cancer patients compared to normal tissue samples [45]. The up-regulated *UPP1* expression increased lymph node metastasis risk and promoted tumor growth. Noushmehr et al. identified hypermethylation of *UPP1* gene in patients belong to the proneural subgroup that were diagnosed with low grade gliomas and many of the patients were classified to have CIMP-positive tumors [17]. Wang et al. found *UPP1* gene expression was up-regulated as glioma grade increased using more than 900 samples from the TCGA database [46]. The Gene Ontology analysis revealed that *UPP1* is likely associated with immune and inflammatory response and the increase of expression negatively impacted patient survival. The extensive data have proven *UPP1* is an efficient prognosis marker for cancer. Our analysis aligns with the previous studies where *UPP1* gene expression was greatest in glioma Grade IV samples with demethylation at the CpG site captured by cg16270885 and correlates with lower patient survival.

All ten prognostic methylation CpG candidates showed good prognostic capability individually but we wanted to investigate their collective efficiency in the prediction of patient survival. We took the poor prognosis conditions obtained from the ten candidates (Table 2 and Figure 6) and assigned patients from the training sample set that met all ten poor survival criteria to the poor prognosis group. The survival probability of the poor prognosis group dropped remarkedly; 50% survival probability at less than 14 months. The good prognosis group had a significantly better survival where the median survival time was more than 81 months. This set of prognostic markers was validated collectively in the validation sample set and a similar trend was observed: median survival time was approximately 18 and 74 months for poor and good prognosis group, respectively.

We plotted the identified prognostic CpG sites and observed a consistent methylation change as glioma grade increases. CpG site of *UPP1* demethylate as glioma grade increases while the remaining genes, *SMOC1, KCNA4,* and *SLC25A21*, methylate along with higher tumor grade. This indicates the methylation pattern changes at these specific CpG sites change collectively as the tumor progresses. We compared the changes at those CpG sites with other molecular alterations in glioma and as well as molecular subtypes assigned by the Ceccarelli et al. We found that the methylation profile of these CpG sites align with specific molecular subtypes. The greatest methylation signal correlation for our probes was observed in samples with IDH mutation and 1p/19 codeletion, with a slight decrease in methylation for samples containing only IDH mutation, and the lowest methylation for samples with IDH wild type genotype and no 1p/19q codeletion. IDH mutation can promote CIMP in gliomas [47] which explains the increased methylation in *UPP1*-associated probes, but our observation suggests that *UPP1* methylation can also be associated with 1p/19q codeletion.

We investigated the sample distribution in both good prognostic and poor prognostic groups to understand the significance of these genes. All samples in the poor prognostic group have IDH wild type genotype with no 1p/19q codeletion, which aligns with current knowledge that IDH mutation and 1p/19q codeletion usually result in more favorable overall survival compared IDH wild type and non-1p/19q codeletion, independently [15]. The poor prognostic group consists of samples with molecular subtypes of classic-like, mesenchymal-like, and one case from LGm6-GBM. As summarized by Ceccarelli et al., samples from G-CIMP-low and the subtypes mentioned above have poorer patient survival compared to the samples classified as codeletion, G-CIMP-High, or Pilocytic Astrocytomas (PA) subtypes. Our results suggest the genes we identified are generally associated with molecular changes observed in glioma since they identify samples with tumors that display classic gene expression signature (classic-like) and mesenchymal-like instead of the codel (1p/19q codeletion) and CIMP subtypes. Mair et al. have reviewed the patient survival data of gliomas and suggest oligodendroglioma has the most favorable survival in Grade II and III gliomas with a median overall survival for oligodendroglioma of at least above 11 years [15]. However, our prognostic model allows us to identify oligodendrogliomas that exhibit similar methylation profile at these CpG sites, as the Grade III astrocytoma, Grade III oligodendrogliomas, and Grade IV GBM, thus refining the prognosis for these patients. We found that most of the poor prognostic samples contain chromosome 7 amplification and chromosome 10 deletion alteration. Chromosome 7 amplification is linked to increased mesenchymal gene expression which supports our findings of these group of CpG sites correctly diagnosing poor prognosis.

We identified the median survival time of the patients from the poor prognosis group to be between 14 to 18 months which is equivalent to the median survival of high glioma grade, glioblastoma multiforme. Interestingly, the patient composition of the poor prognosis group was a mixture of mainly glioma Grade III and glioma Grade IV patients suggesting the devastating effect is observed across tumor grades.

## 4. Materials and Methods

### 4.1. Data Sources

DNA methylation data of the 682 glioma tissue samples were downloaded from the TCGA database under LGG (532 samples) and GBM projects (150 samples) through the TCGABiolinks R package (version 3.12; https://bioconductor.org/packages/release/bioc/html/TCGAbiolinks.html) in June 2020 [48,49,50]. Methylation data, captured through the Illumina Infinium HumanMethylation 450 platform, is expressed in β-value, which is the estimated methylation level calculated from the methylation intensity over the methylation and unmethylation intensities. The data were separated into a training and a validation set. Among the 400 training set samples, 349 samples have gene expression quantification data available in the TCGA database and they were downloaded through the TCGABiolinks R package in November 2020. Gene expression data are expressed as Fragments Per Kilobase Million (FPKM). Patient’s clinical data for the 682 glioma tissue samples included age, gender, survival information, histological grade, and histological subtype were obtained from the portal of FireBrowse (http://firebrowse.org/) in June 2020. The clinicopathological characteristics of patients in the training sample set and validation sample set can be found in Table 1.

### 4.2. Methylation Data Manipulation

We analyzed methylation data around CpG sites located on the CpG island, CpG island shore (up to 2000 bp upstream and downstream from CpG island), and CpG island shift (up to 4000 bp upstream and downstream from CpG island). CpG sites that contain known single nucleotide polymorphisms, obtained from the Illumina product support website (https://support.illumina.com/array/array_kits/infinium_humanmethylation450_beadchip_kit), within two base pairs of the targeted CpG site and cross-reactive sites were removed from the data set to reduce potential false interpretation of the methylation state [51,52]. Lastly, we removed CpG sites on the Y chromosome as no data shall be available for female patients and will eventually be filtered out before the analysis. The above filtering steps reduced the data set down to 180,758 CpG sites for differential variability analysis.

### 4.3. Differential Variability Analysis

We used differential variability analysis to identify CpG sites with a significant change in methylation variance observed in one group while the contrasting group displays a consistent methylation level. The two comparisons performed in the study were Glioma Grade II compared to Glioma Grade III and Glioma Grade III compared to Glioma Grade IV. The analyses were performed using the limma R package (version 3.42.2; https://www.bioconductor.org/packages/release/bioc/html/limma.html) and missMethyl R package (version 1.20.4; http://bioconductor.org/packages/release/bioc/html/missMethyl.html) [53,54,55]. The variability of each CpG site is calculated followed by a fitted linear model. CpG sites with FDR ≤ 0.05 from the differential variability analysis output are considered DV CpG sites. All DV CpG sites from the Grade II and Grade III comparison and the top 25% DV CpG sites from the Grade III and Grade IV comparison were inputted for methylation module clustering.

### 4.4. Weighted Correlation Network Analysis (WGCNA)

We used the WGCNA R Package (version 1.69; https://cran.r-project.org/web/packages/WGCNA/index.html) to cluster correlated DV CpG sites within the data set [26]. In the analysis, a soft thresholding power (β) of 3, where the R^2^ of the fitted scale-free topology model at 0.9 is met, was selected for network construction. The topology overlap matrix was used to calculate the interconnectivity between two CpG sites and construct comethylation modules with a minimum of 100 CpG sites per module. The identified comethylation modules were then correlated with the clinical trait of interest for this study: glioma grade. Modules with module-trait correlation ≥ |0.5| and *p*-value ≤ 0.05 were selected. We narrowed output to CpG sites by including only sites with CpG site significance ≥ 0.5 and *p*-value ≤ 0.05 from the highly correlated modules. The CpG site significance of the CpG site represents the absolute value of the correlation between methylation expression of the CpG site and the glioma grade.

### 4.5. Gene Expression Data

Gene expression data are mapped to Emsembl IDs while methylation data were annotated with gene symbols. We used the ensembldb R Package (version 3.12; https://www.bioconductor.org/packages/release/bioc/html/ensembldb.html) to convert between Emsembl ID and gene symbol [56].

### 4.6. Survival Analysis

Survival analysis was performed on the selected CpG sites with the associated patient survival information. We used the survival R package (version 3.2.7; https://cran.r-project.org/web/packages/survival/index.html) alongside with survminer R package (version 0.4.8; https://cran.r-project.org/web/packages/survminer/index.html) for visualization [57]. The survival curve is fitted through the Kaplan-Meier plot with a *p*-value calculated from the log-rank test to evaluate the significance of the survival prediction. Using the group median methylation level as the cutoff, patients with β-value greater than the cutoff at the interested CpG site were assigned to the hypermethylation group while the patients with β-value less than or equal to median were classified as the hypomethylation group.

## 5. Conclusions

Glioblastoma multiforme is known to be an aggressive brain tumor with a median survival of 15 to 16 months while Grade II and Grade III gliomas are less destructive and leave patients with longer survival time. However, lower-grade gliomas will ultimately progress to glioblastoma. We need improved systematic approaches to monitor tumor progression and predict patient survival based on the real-time molecular changes that result from tumorigenesis and progression. The increased attention around tumor detection through somatic mutations found in cellular tumor DNA permits an early and noninvasive way for diagnosis [58]. Methylation changes identified via methylation sequencing provide more context to the diagnostic process and patient care optimization by pinpointing the origin of cancerous sites as well as providing progression monitoring [59].

In conclusion, we identified ten methylation markers affecting four genes (*SMOC1*, *KCNA4*, *SLC25A21*, and *UPP1*) that are associated with glioma grade progression, and demonstrate a strong prognostic probability for patient prognosis which is also able to identify patients usually considered to be good survivors (oligodendroglioma). We found the gene expression level of *SMOC1, KCNA4, SLC25A21,* and *UPP1* to be closely correlated with the methylation level of specific CpG sites for each gene. The methylation signal of these ten CpG sites changed progressively with glioma grade and they showed good prognostic capability collectively. In addition, the identified CpG sites show high correlation with molecular subtype, IDH alteration, and chromosome 1p/19q alteration, strengthening the validity of our model.

Our ten-methylation-marker model predicts survival for patients with oligodendrogliomas that exhibit similar epigenetic profiles as patients with higher grade glioma and poorer prognosis. As there is some evidence suggesting that *UPP1* and *SMOC1* are markers for glioma, our finding of *KCNA4* and *SLC25A21* add to the previously identified gene list to fortify patient outcome prediction and guide future investigations on the impact of these genes and their pathways involved in glioma progression. The ten methylation markers evaluated in this study will contribute to the continuous improvement of patient prognosis; patient prognosis should imitate the model of precision medicine where patients are treated based on their unique circumstances and given precise diagnosis to receive the appropriate medical care.

## Figures and Tables

**Figure 1 ijms-22-01020-f001:**
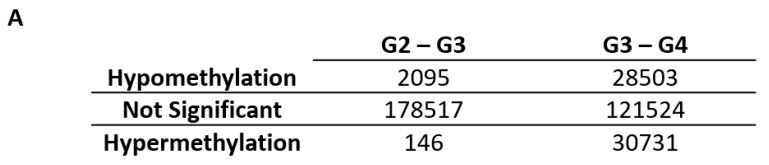
Differential variability analysis of training data. (**A**) Summary of differential variability analysis. The numbers of significant hypo- and hypermethylated Cytosine-phosphate-Guanine (CpG) sites in each comparison are listed. Differential variability analysis of training data. (**B**) Heatmaps presenting the methylation status of differentially variable (DV) CpG sites identified through differential variability analysis conducted on the training data sample. Left panel shows all the DV CpG sites from Grade II and Grade III comparison and the right panel shows a subset of DV CpG sites from Grade III and Grade IV comparison. Glioma samples are arranged by grade.

**Figure 2 ijms-22-01020-f002:**
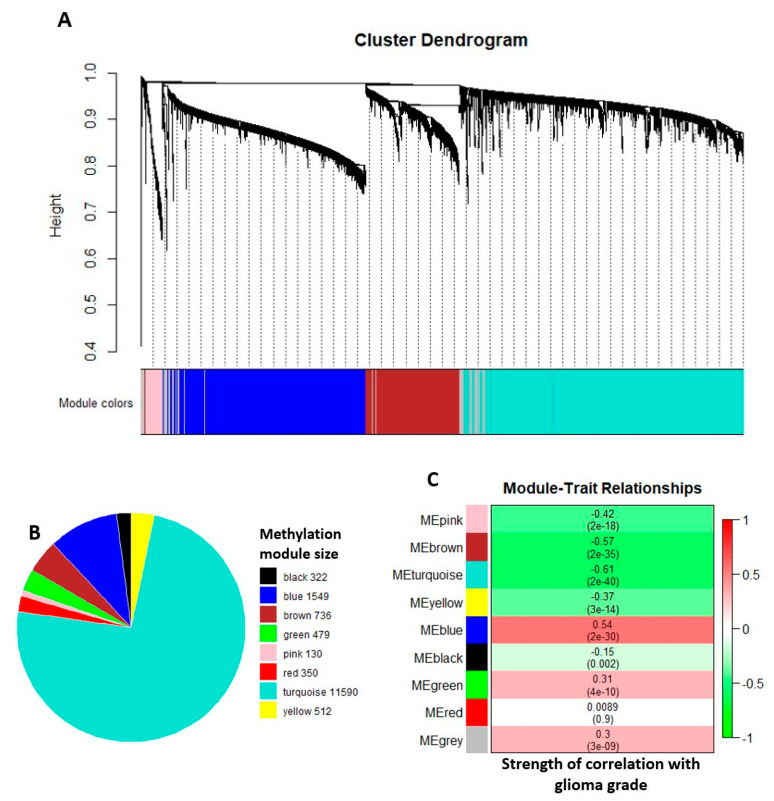
Weighted Correlation Network Analysis (WGCNA) output of the training data set. (**A**) The hierarchical clustering dendrogram of approximately 5000 DV CpG sites inputted to WGCNA. This is a representative cluster block of the four blocks used in network construction. (**B**) The WGCNA identified 8 methylation modules and they are represented by different colors in the pie chart. The grey module is excluded from the pie chart due to CpG sites falling within the grey modules have low correlation with the other methylation modules. (**C**) Module and clinical trait association. Correlation between module and clinical trait (glioma grade) is calculated and displayed on the heatmap, with the associated *p*-value. The relationship is demonstrated by color; red to green where green denotes negative correlation while red denotes positive correlation.

**Figure 3 ijms-22-01020-f003:**
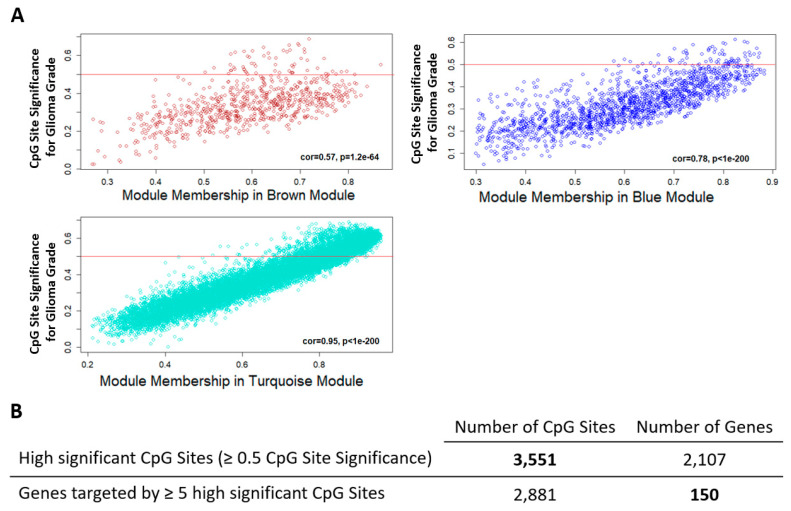
Identification of significant CpG sites. (**A**) Scatterplots of CpG site significance (y-axis) vs. module membership (x-axis) in the three glioma grade-associated methylation modules. Overall CpG site significance and Intramodular connectivity correlation is denoted as cor in the plot. The red line on the plot represents the assigned cutoff at |0.5| significance of the correlation of the CpG site with trait of interest. (**B**) Summary of significant glioma tumor grade-associated CpG sites outputted from WGCNA.

**Figure 4 ijms-22-01020-f004:**
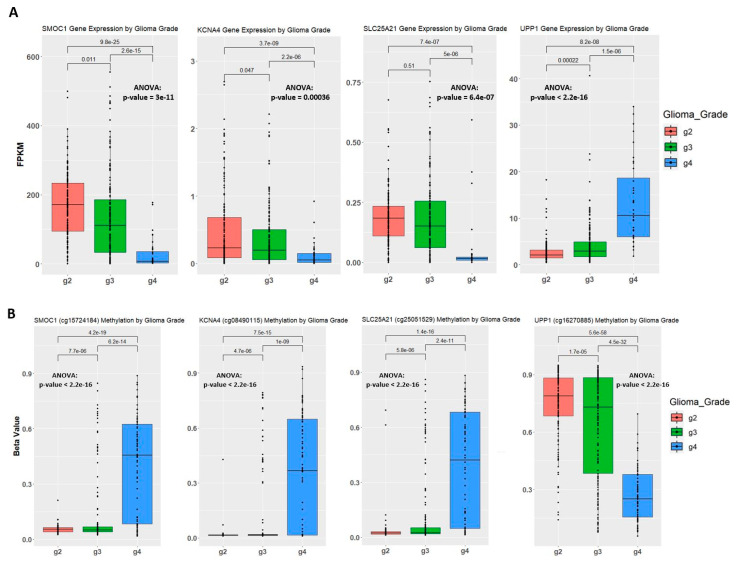
Correlation between gene expression and methylation level in the training set. (**A**) Gene expression of samples by glioma grade. Fragments Per Kilobase Million (FPKM) stands for Fragments Per Kilobase of transcript per Million mapped reads. The t.test method is used for statistical analysis when comparing two groups of samples. One-way analysis of variance (ANOVA) was used when comparing three groups of samples on one variable. (**B**) Methylation level of samples by glioma grade. Methylation level is represented by beta-value. One probe is selected to show the methylation level for *SMOC1* gene and *KCNA4* gene. (**C**) Boxplots of the *IGF2BP3* and *Fn14* gene expression and methylation level on one of the associated probes. Kaplan-Meier plot measuring patient survival through the methylation level of CpG sites captured by cg00508334 for *IFG2BP3* and cg00510447 for *Fn14*.

**Figure 5 ijms-22-01020-f005:**
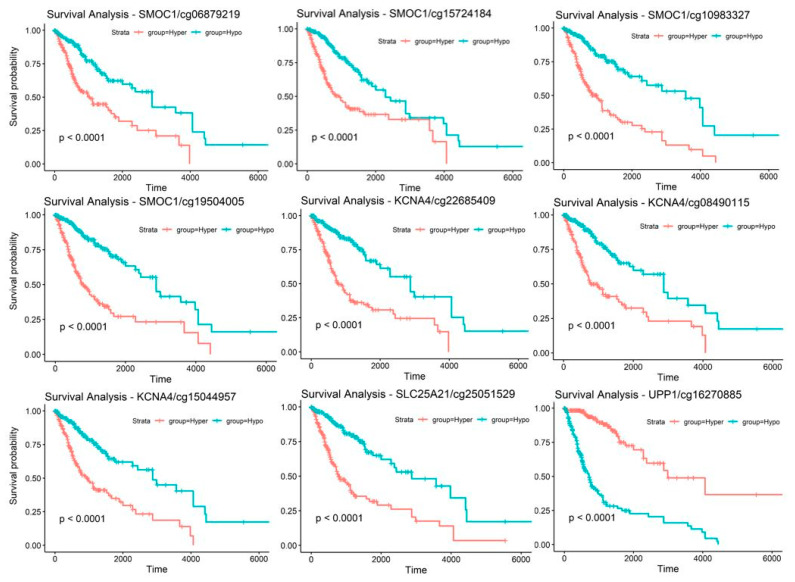
Kaplan-Meier plots measuring patient survival through the methylation level of each CpG sites. Glioma patients are separated into hypermethylation group or hypomethylation group. Nine out of the ten prognosis markers are shown in this figure. The excluded prognosis marker is cg08754456 that targets *SMOC1* gene and it shows a similar survival curve as the rest of the prognosis markers from *SMOC1* gene.

**Figure 6 ijms-22-01020-f006:**
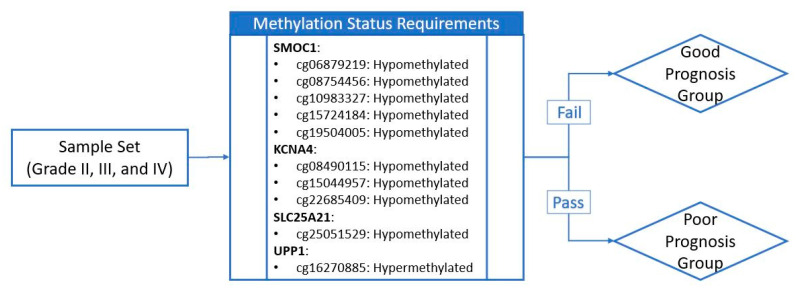
Schematic diagram of sample grouping criteria. If the sample meets methylation status requirements for all ten CpG sites (represented by probe IDs from the array product design), it is assigned to the poor prognosis group. If the sample fails one or more of the methylation status requirements, it is assigned to the good prognosis group. Hypo- and hypermethylation status of the sample is determined through comparison with the group median methylation level.

**Figure 7 ijms-22-01020-f007:**
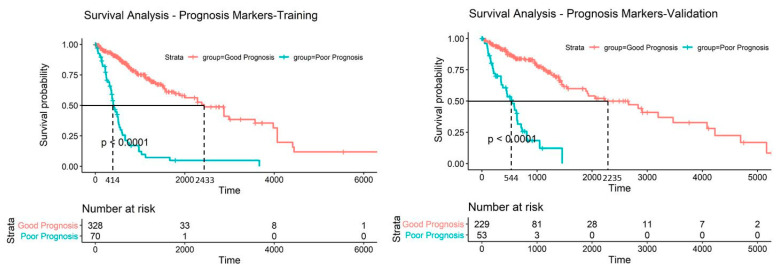
Prognosis of patient survival through methylation level of the ten CpG sites. Kaplan-Meier plots of the glioma patients separated into good prognosis group and poor prognosis group. 50% survival is between approximately 14 months and 18 months in the training set and validation set, respectively. *p*-values for both data sets are less than 0.0001.

**Figure 8 ijms-22-01020-f008:**
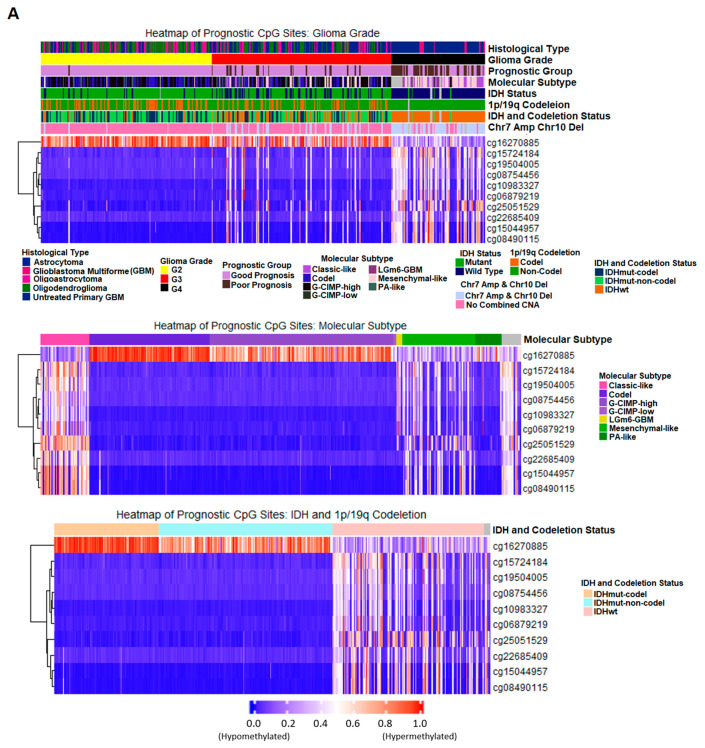
Heatmap of prognostic markers and sample distribution from training sample set. (**A**) Heatmaps displaying the identified ten CpG sites for the training sample set sorted by Glioma Grade, molecular subtype, and isocitrate dehydrogenase (IDH) with 1p/19q codeletion alterations in the order of top, middle, and bottom. Light grey on the top of the status bar denotes NA for samples with no information given for the specific status. (**B**) Bar charts of training sample distribution in the good prognosis and poor prognosis group with additional molecular and/or clinical data presented. NA indicates no information available for the associated molecular alteration. Left: Molecular subtype distribution in sample groups with different IDH and 1p/19q codeletion genotypes within the good and poor prognosis groups. Middle: Histological type distribution in different glioma grade sample groups within the good and poor prognosis groups. Right: Molecular subtype distribution in samples groups with different CNA (copy number alteration) on chromosome 7/10 within the good and poor prognosis groups. Chromosome 7 amplification and chromosome 10 deletion: abbreviated as Chr7/10. Number of samples from each molecular and clinical subtype can be found in Table 3.

**Table 1 ijms-22-01020-t001:** Clinicopathological data of training sample set and validation sample set.

	Training (400 Samples)		Validation (282 Samples)	
	Number of Samples	Percent of Samples	Number of Samples	Percent of Samples
Age	46.19 yo	N/A	46.69 yo	N/A
Days to death	888.7 days	N/A	916.6 days	N/A
Days to last follow up	834.9 days	N/A	866.4 days	N/A
Grade II	154	38.5%	107	37.9%
Grade III	162	40.5%	109	38.7%
Grade IV	84	21.0%	66	23.4%
Astrocytoma	119	29.7%	78	27.7%
Oligoastrocytoma	80	20.0%	55	19.5%
Oligodendroglioma	117	29.3%	83	29.4%
Glioblastoma multiforme (GBM)	84	21.0%	66	23.4%
Alive	259	64.8%	182	64.5%
Dead	141	35.2%	100	35.5%

**Table 2 ijms-22-01020-t002:** Summary of Prognostic CpG Sites Associated with Glioma Tumor Progression.

Probe ID	CpG Position Relative to Targeting Gene	Locus	Gene	CpG Position Relative to CpG Island	Methylation Status for Poor Prognosis *
cg16270885	5’UTR **	chr7:48096722	UPP1	S_Shore †	Hypomethylation
cg08490115	TSS200 ‡	chr11:29995251	KCNA4	Island	Hypermethylation
cg15044957	TSS200	chr11:29995248	KCNA4	Island	Hypermethylation
cg22685409	TSS1500 ††	chr11:29995268	KCNA4	Island	Hypermethylation
cg06879219	TSS200	chr14:69415835	SMOC1	Island	Hypermethylation
cg08754456	TSS200	chr14:69415812	SMOC1	Island	Hypermethylation
cg10983327	TSS200	chr14:69415825	SMOC1	Island	Hypermethylation
cg15724184	1st Exon	chr14:69416230	SMOC1	Island	Hypermethylation
cg19504005	1st Exon	chr14:69416170	SMOC1	Island	Hypermethylation
cg25051529	Body	chr14:36711213	SLC25A21	Island	Hypermethylation

* Methylation status for poor prognosis compared to group median methylation level. ** Within 5’ untranslated region. † 0–2 kb downstream (3’) of CpG island. ‡ 0–200 bases upstream of the transcriptional start site. †† 200–1500 bases upstream of the transcriptional start site

**Table 3 ijms-22-01020-t003:** Summary table of the patient distribution in different prognosis group for training set and validation set with clinical and molecular information.

	Training Set (400)		Validation Set (282)	
	Good Prognosis (330)	Poor Prognosis (70)	Good Prognosis (229)	Poor Prognosis (53)
**Clinical**				
*Grade*				
Grade II	153	1	104	3
Grade III	141	21	95	14
Grade IV	36	48	30	36
*Histology*				
Astrocytoma	106	13	69	9
Oligodendroglioma	115	2	78	5
Oligoastrocytoma	73	7	52	3
Glioblastoma	36	48	30	36
**Molecular**				
*Isocitrate Dehydrogenase (IDH)*				
Mutant	256	0	185	0
Wild Type	72	67	39	52
NA	2	3	5	1
*1p/19q codeletion*				
Codeletion	96	0	76	0
No codeletion	233	70	153	51
NA	1	0	0	2
*IDH and 1p/19q codeletion status*				
IDH mutant and codeletion	96	0	76	0
IDH mutant and no-codeletion	159	0	109	0
IDH wild type and no-codeletion	72	67	39	50
NA	3	3	5	3
*Chr7 Amplification/Chr 10 Deletion*				
Chr7 Amp/Chr 10 Del	40	55	19	40
No combined copy number alteration	288	13	206	9
NA	2	0	4	4
*Molecular Subtype (Ceccarelli et al., 2016)*				
Classic-like *	8	33	6	31
Codel	100	0	76	0
G-CIMP **-high	153	0	97	0
G-CIMP **-low	2	0	10	0
LGm6-GBM	4	1	7	1
Mesenchymal-like	38	23	19	20
Pilocytic Astrocytomas (PA) like	22	0	7	0
NA	3	3	7	1

* Classic-like: classical gene expression signature in tumor. ** CIMP: Cytosine-phosphate-Guanine Island Methylator Phenotype.

## Data Availability

Access to the data presented in this study are available in the Materials and Methods section and in the Appendix A section.

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
