# Peer review of "DNA Methylation Analysis Identifies Patterns in Progressive Glioma Grades to Predict Patient Survival"

_ijms, 2021, doi:10.3390/ijms22031020_

Round 1

Reviewer 1 Report

Weng and Salazar have used data available through The Cancer Genome Atlas database to identify methylation patterns of CpG sites that stratify patients into good and poor prognosis groups. Three of the four highlighted genes (SMOC1, KCNA4, SLC25A21) have not been previously linked to glioma prognosis and the fourth (UPP1) only recently.

Below are suggestions for improvement of the manuscript:

Page 1, last paragraph: The WHO Classification of CNS tumours describes more than 20 subtypes of glioma, not just 4. It would be better to simply say: “Gliomas are classified based on the World Health Organisation…”

Figure 1: This figure is not very clear. It would be helpful if there was some indication of which sites are in which part of the heatmap eg which part of the figure is the sites that distinguish grade 2 & 3 tumour and which part the sites that distinguish grade 3 & 4. If there are other distinguishing characteristics that led to inclusion of particular groups of sites in the subset, it would be good to indicate those. Given that the grade 3 & 4 tumours are separated by nearly as many hypomethylated sites as hypermethylated sites, I would expect the Grade 4 section of the heatmap to be nearly half red. Why is it nearly all blue?

Figure 2C: Rather than “glioma grade” the chart is colour coded to reflect “Strength of correlation with glioma grade”

Discussion: It would be interesting to have some discussion (or even see results) of how well stratification by methylation status predicts survival compared to stratification by grade and whether high grade patients with favourable methylation have similar survival to low grade with favourable methylation or fall between the two groups.

Out of interest, UPP1 has been recently reported by others to be a prognostic indicator in glioma (Wang, Xu et al https://www.ncbi.nlm.nih.gov/pmc/articles/PMC7433823/)

Typographic and grammatical corrections:

Page 1, line 37-38 tumour suppressor geneS

Results Page 2, line 83: FRD – I think this is a typo for FDR,
line 87: FDR needs definition at first use.
line 92-93: Figure 1B is the table and 1A is the heatmap
Page 3, line 104: DV needs definition at first use.

Page 4, line 112: module size showN on the legend

Figure 2 legend: The grey module is excluded from the pie chart AS CpG sites FALLING within…

Page 5, line 142: high correlation between gene significance and module membership indicatING the CpG sites…

Author Response

Response to Reviewer 1 Comments

We thank Reviewer 1 for the constructive feedback.

Point 1: Page 1, last paragraph: The WHO Classification of CNS tumours describes more than 20 subtypes of glioma, not just 4. It would be better to simply say: “Gliomas are classified based on the World Health Organisation…”

Response 1: We thank Reviewer 1 for the clarification. We changed the wording as indicated by the reviewer.

Point 2: Figure 1: This figure is not very clear. It would be helpful if there was some indication of which sites are in which part of the heatmap eg which part of the figure is the sites that distinguish grade 2 & 3 tumour and which part the sites that distinguish grade 3 & 4. If there are other distinguishing characteristics that led to inclusion of particular groups of sites in the subset, it would be good to indicate those. Given that the grade 3 & 4 tumours are separated by nearly as many hypomethylated sites as hypermethylated sites, I would expect the Grade 4 section of the heatmap to be nearly half red. Why is it nearly all blue?

Response 2: We thank Reviewer 1 for the suggestions. We have now separated the heatmap into two in Figure 1B line 124. One heatmap includes all the DV CpG sites identified from the Grade II and Grade III comparison while the other heatmap shows the top 10% DV CpG sites (selected based on the lowest FDR value) from Grade III and Grade IV comparison. There were no other distinguishing characteristics that led to inclusion of particular groups of sites in the subset, other than the cutoff for FDR ≤ 0.05. This is a very good point the reviewer brings up. While we cannot say for certain because of our limited capacity to plot all the methylation sites in one picture, we can speculate that for the limited number of sites plotted (n = 5923) the hypomethylation phenotype for the subset of CpG sites were considered more significant changes between the comparison groups due to the lower FDR value. We have clarified it in lines 113-115 Despite this interesting point, are findings are consistent with expected hypomethylation association with tumor progression, as noted in the discussion section, lines 317-319.

Point 3: Figure 2C: Rather than “glioma grade” the chart is colour coded to reflect “Strength of correlation with glioma grade”

Response 3: We thank Reviewer 1 for the suggestion. We have relabelled Figure 2C to change “Glioma Grade” to “Strength of correlation with glioma grade”.

Point 4: Discussion: It would be interesting to have some discussion (or even see results) of how well stratification by methylation status predicts survival compared to stratification by grade and whether high grade patients with favourable methylation have similar survival to low grade with favourable methylation or fall between the two groups.

Response 4: We thank Reviewer 1 for the suggestion. We have included other glioma-associated molecular alteration as well as molecular subtype as indicated by Ceccarelli et al. (Cell, 2016, 164, 550–563, doi: 10.1016/j.cell.2015.12.028) in our study since the understanding molecular alteration is important to consider for patient outcome. These extensively evaluated and accepted molecular parameters stratify patients better and exhibit better prognosis probability. Therefore, we have included the molecular alteration and subtype in our study.

These results are described and discussed in section 2.7 and 3, page 11-13, 15, lines 268-308, 403-438, Figure 8 and Table 3. The identified loci correlate well with the molecular subtype (result illustrated in figure 8A) and the poor survival prognosis corresponds to the classic-like (classic gene expression profile in tumor) and mesenchymal-like subtypes that are known to result in worse patient survival. In addition, we found the methylation signal of identified CpG sites is highly related to IDH alteration while UPP1-associated CpG also closely relates to chromosome 1p/19q codeletion.

While our model of CpG loci is not necessarily better at predicting survival than the currently established brain tumor molecular subtype classification when all glioma molecular alterations are included in the evaluation, our findings identified patients that would otherwise be classified as good survivors (oligondedroglioma patients).

Point 5: Out of interest, UPP1 has been recently reported by others to be a prognostic indicator in glioma (Wang, Xu et al https://www.ncbi.nlm.nih.gov/pmc/articles/PMC7433823/)

Response 5: We thank Reviewer 1 for the suggestion. We added the reference in line 387, (reference #46).

Point 6: Typographic and grammatical corrections:

Page 1, line 37-38 tumour suppressor geneS

Results Page 2, line 83: FRD – I think this is a typo for FDR,
line 87: FDR needs definition at first use.
line 92-93: Figure 1B is the table and 1A is the heatmap
Page 3, line 104: DV needs definition at first use.

Page 4, line 112: module size showN on the legend

Figure 2 legend: The grey module is excluded from the pie chart AS CpG sites FALLING within…

Page 5, line 142: high correlation between gene significance and module membership indicatING the CpG sites…

Response 6: We thank Reviewer 1 for the suggestions. All typographic and grammatical corrections were all made as indicated and marked by track changes.

Reviewer 2 Report

Some minor points have to be corrected.

  1. Please try to use more references from the last 5 years (from 2016)
  2. While the work is focused on methylation, a brief introduction paragraph would be advisable on other epigenetic mechanisms for regulating gene expression.
  3. Minor linguistic errors should be eliminated.

Author Response

Point 1: Please try to use more references from the last 5 years (from 2016).

Response 1: We thank Reviewer 1 for the feedback. With the additional paragraph and other noted changes, the introduction now provides sufficient background and include all relevant references.

The following references since 2016 have been added to the manuscript:

    1. Nature, 2016, 530, 441–446, doi: 10.1038/nature16998. Line number: 36 (reference number 20)
    2. Genes, 2019, 10:257, doi: 10.3390/genes10040257. Line number: 30 (reference number 3)
    3. Cancer Letters, 2017, 396:130-137, doi: 10.1016/j.canlet.2017.03.029. Line number: 35 (reference number 6)
    4. International Journal of Molecular Sciences, 2018, 19:1166, doi: 10.3390/ijms19041166. Line number: 35 (reference number 7)
    5. Epigenetics, 2018, 13:473-489, doi: 10.1080/15592294.2018.1469894. Line number: 87 (reference number 20)
    6. F1000Research, 2019, 8:2106, doi: 10.12688/f1000research.21584.2. Line number: 87 (reference number 21)
    7. Cancer Gene Therapy, 2020, 27:702–714, doi: 10.1038/s41417-019-0142-6. Line number: 88 (reference number 23)
    8. Frontiers in Genetics, 2019, 10:786, doi: 10.3389/fgene.2019.00786. Line number: 88 (reference number 24)
    9. Oncology Letters, 2019, 18:5831-5842, doi: 10.3892/ol.2019.10931. Line number: 88 (reference number 25)

Point 2: While the work is focused on methylation, a brief introduction paragraph would be advisable on other epigenetic mechanisms for regulating gene expression.

Response 2: We have now included a brief introduction paragraph on other epigenetic mechanism for regulating gene expression. The paragraph is on page 1-2 and lines 40-58.

Point 3: Minor linguistic errors should be eliminated.

Response 3: We proofread the article and eliminated all minor linguistic errors.

Reviewer 3 Report

The manuscript by Jing Yin Weng and Nicole Salazar, entitled “DNA Methylation Analysis Identifies Progressive Methylation Patterns in Accordance with Increased Glioma Grade to Predict Patient Survival”, deals with the discovery of a relationship between DNA methylation, glioma malignancy and patient survival.
Through an extensive analysis of gene methylation patterns carried out in glioma tissue samples downloaded from The Cancer Genome Atlas database under Low Grade Gliomas (532 samples) and Glioblastoma multiforme (150 samples) projects, the Authors selected four genes, (namely, SMOC1, KCNA4, SLC25A21, and UPP1) where ten CpG sites shown significant changes in expression (hyper- or hypo-expression) and methylation (hyper- or hypo-methylation) levels, which were correlated with glioma grade and patient survival.
Proteins encoded by the four selected genes are involved in cancer, one of them (SMOC1) with a role which may be related to glioma grade.
In the lack of any evidence of the actual role of the selected genes in gliomas, the importance of these findings remains marginal. For instance, it cannot be ruled out that the poor survival prognosis predicted for patients whose genoma met all ten poor survival criteria, though suggestive, might be due to other factors, which had not been analysed in the present study.
In this respect, the Authors reported that another study, which previously analysed the CpG island methylation phenotype (CIMP) in samples from The Cancer Genome Atlas database for different glioma grades, found several hypermethylations on the CpG island of a subset of genes (ANKRD43, HFE, MAL, FAS-1,etc.) and classified glioma samples as CIMP-positive or CIMP-negative based on the methylation level detected on those genes. This study already established a correlation between CIMP-positive tumors and better survival.

Therefore, apart from the identification of additional genes which grow the list of altered genes involved in gliomas, the advance provided by the present study remains limited. The Authors should address in what their data are innovative respect to these previous findings, and in what the selected genes they found are more predictive of patient survival compared with the genes already identified by the previous study.

Author Response

Point 1: In the lack of any evidence of the actual role of the selected genes in gliomas, the importance of these findings remains marginal.

Response 1: We thank Reviewer 2 for the feedback and constructive suggestions. With the additional analysis and considerations as noted on the lines below, our results and discussion have improved significantly.

To respond to the comment: we verified the role of the identified genes in cancer. UPP1 has been identified as hypermethylated, mainly in low grade gliomas, and our findings align with the previous study (reference: Cancer cell, 2010, 17, 510–522, doi: 10.1016/j.ccr.2010.03.017.) We have noted this in our discussion in lines 374-376. We note in the discussion in lines 328-332 that SMOC1 may have a role which may be related to glioma grade (reference: BMC cancer, 2004, 4, 39, doi: 10.1186/1471-2407-4-39). KCNA4 was studied in gastric cancer where hypermethylation of the promoter serves as a good detection marker (reference: Clinical biochemistry, 2011, 44, 1405–1411, doi: 10.1016/j.clinbiochem.2011.09.006, lines 345-348). Many members of the SLC25 family were studied in multiple cancer types including lung and breast cancer (lines 358-361) suggesting the potential of SLC25A21 as a promising candidate. Although KCNA4 and SLC25A21 were not previously associated with gliomas, the successful identification of UPP1 and SMOC1 suggests our approach is useful for identifying undiscovered glioma-associated methylation loci.

Point 2: For instance, it cannot be ruled out that the poor survival prognosis predicted for patients whose genoma met all ten poor survival criteria, though suggestive, might be due to other factors, which had not been analysed in the present study.

Response 2: We thank Reviewer 2 for the constructive suggestions. We have now included other factors in our analysis to better understand the biological relationship between our identified CpG methylation loci and glioma WHO grades 2, 3, and 4 which has not been explicitly established in the literature in section 2.7 and 3, page 11-13, 15, lines 259-299, 394-429, Figure 8 and Table 3. Specifically, we have now included molecular subtypes including IDH status, copy number alteration on chromosome 7 and 10 as well as 1 and 19, histological subtypes, molecular subtype as indicated by Ceccarelli et al. (Cell, 2016, 164, 550–563, doi: 10.1016/j.cell.2015.12.028). The identified loci correlate well with the molecular subtype (result illustrated in figure 8A) and the poor survival prognosis corresponds to the classic-like (classic gene expression profile in tumor) and mesenchymal-like subtypes that are known to result in worse patient survival. In addition, we found the methylation signal of identified CpG sites is highly related to IDH alteration while UPP1-associated CpG also closely relates to chromosome 1p/19q codeletion.

Point 3: In this respect, the Authors reported that another study, which previously analysed the CpG island methylation phenotype (CIMP) in samples from The Cancer Genome Atlas database for different glioma grades, found several hypermethylations on the CpG island of a subset of genes (ANKRD43, HFE, MAL, FAS-1,etc.) and classified glioma samples as CIMP-positive or CIMP-negative based on the methylation level detected on those genes. This study already established a correlation between CIMP-positive tumors and better survival.

Response 3: We thank Reviewer 2 for the constructive suggestion. We have now included molecular subtypes, where samples are additionally classified as G-CIMP-high or G-CIMP-low, to understand the relationship between our methylation CpG sites and CIMP in section 2.7 and 3, page 11-13, 15, lines 259-299, 394-429, Figure 8 and Table 3. While CIMP analysis looks at promoter methylation regions, our analysis looks at specific loci not restricted to promoter regions. CIMP analysis does not distinguish between different glioma grades, whereas our analysis focuses on identifying loci that change in methylation signal as glioma grade increases and in addition predicts patient survival using the model established in our study. 

Point 4: Therefore, apart from the identification of additional genes which grow the list of altered genes involved in gliomas, the advance provided by the present study remains limited. The Authors should address in what their data are innovative respect to these previous findings, and in what the selected genes they found are more predictive of patient survival compared with the genes already identified by the previous study.

Response 4: We thank Reviewer 2 for the constructive suggestion.

Our identification of a novel combination of 10 methylation probes for SMOC1, KCNA4, SLC25A21 combined with the UPP1 probes, enable monitoring tumor progression from WHO grade 2 to grade 3 to grade 4 and make for good vs poor overall survival predictions. The concurrence of our methylation probes with glioma molecular subtypes and glioma specific molecular alterations like IDH and chromosome 1 and 19 codeletion confirms the robustness of our findings in identifying glioma relevant methylation probes. While our model of CpG loci is not necessarily more predictive than the currently established brain tumor molecular subtype classification when all glioma molecular alterations are included in the evaluation, our findings identified two genes that were not previously studied in glioma (KCNA4 and SLC25A21). The UPP1 and SMOC1 identified along with KCNA4 and SLC25A21 suggests these latter two genes are relevant to glioma since they exhibit similar methylation patterns and together, identified poor prognosis samples that belong almost exclusively, to the classic-like and mesenchymal-like subtypes out of seven different glioma molecular subtypes.

These methylation CpG sites could serve as marker sets to help better stratify patients that would otherwise be considered to have good prognosis, when in reality, they could actually be poor survivors. Our model assists in identifying poor prognosis oligodendroglioma patients (Figure 8B and Table 3), because they exhibit similar methylation profiles as the more invasive and/or higher-grade gliomas, despite oligodendroglioma patients typically being classified with good survival.

Our data therefore are innovative because the identified CpG sites and genes are candidates for continued study and improved targeted therapies leading to precision medicines that could reduce mortality, even if for a limited subgroup of brain tumor patients, as we see for our improved prediction of oligodendroglioma patients. We have clarified this information in section 2.7 and 3, page 11-13, 15, lines 259-299, 394-429, Figure 8 and Table 3, and concluded the finding in section 5, page 17, lines 516-535.

Round 2

Reviewer 3 Report

The Authors fully satisfied the reviewer's request.